



# Lagrangian eddy tracking reveals the Eratosthenes anticyclonic attractor in the eastern Levantine basin

Alexandre Barboni[1,2], Ayah Lazar[3], Alexandre Stegner[1], and Evangelos Moschos[1]

[1]Laboratoire de Météorologie Dynamique, Ecole Polytechnique, 91128 Palaiseau, France
[2]Département de Géosciences, Ecole normale supérieure de Paris, 24 rue Lhomond, 75005 Paris, France
[3]Israel Oceanographic and Limnological Research Center, Hubert Humphry St, Haifa, Israel

**Correspondence:** Alexandre Barboni (alexandre.barboni@polytechnique.edu)

**Abstract.** Statistics of anticyclone activity and trajectories in the southeastern Mediterranean sea over the period 2000-2018 is created using the DYNED atlas, which links the automated mesoscale eddy detection by the AMEDA algorithm with in situ oceanographic observations. This easternmost region of the Mediterranean sea, delimited by the Levantine coast and Cyprus, has a complex eddying activity, which has not yet been fully characterized. In this paper we use Lagrangian tracking
to investigate the eddy fluxes and interactions between different subregions in this area. We find that the southeastern Levantine area is isolated, with no anticyclone exchanges with the western part of the basin. Moreover the anticyclonic structure above the Eratosthenes seamount is identified as being an anticyclone attractor, differentiated from other anticyclones and staying around this preferred position up to four years with successive mergings. Colocalized in situ profiles inside eddies provide quantitative information on their subsurface structure and show that similar surface signatures correspond to very different
physical properties. Despite interannual variability, the so-called "Eratosthenes attractor" stores a larger amount of heat and salt than neighboring anticyclones, in a deeper subsurface anomaly that usually extend down to 500 m. This suggests that this attractor could concentrate heat and salt from this sub-basin, which will impact the properties of intermediate water masses created there.

## 1   Introduction

The circulation in the eastern part of the Mediterranean sea has not been investigated as extensively as the western part, and some aspects of its circulation are still a subject for scientific debate. Different pathways for the mean flow were proposed with notable differences in the Libyan gulf and the Levantine basin (Robinson et al., 1991; Hamad et al., 2006). Since the satellite Sea Surface Temperature (SST) images in the 90's there is an overall agreement on the mean counter-clockwise surface circulation in the eastern Mediterranean basin, the Atlantic waters (AW) coming through the Strait of Sicily, following
the Libyo-Egyptian coast and then continuing along the Levantine and Turkish coasts (Hamad et al., 2006).

The Levantine basin, defined as the part of the eastern Mediterranean east of 23° E and south of 37° N (Hamad et al., 2006), appears to have a rather complex and turbulent circulation, particularly in its southeastern part, bound by the topography of Cyprus and the Egyptian and Levantine coasts. Extensive in situ oceanographic surveys were performed in the previous decades





(Robinson et al., 1991; Brenner, 1993; Hayes et al., 2011), notably with the work of the POEM group, already detecting in the 80's some recurrent large long-lived (lasting longer than a year) anticyclonic structures : Ierapetra south-east of Crete, Mersa-Matruh offshore Egypt, and the Cyprus eddy south of Cyprus (see notably scheme from Robinson et al. (1991)). South of Cyprus, different authors proposed a multi-pole structure named "Shikmona", of which they named the most active feature "the Cyprus eddy" (Brenner, 1993; Zodiatis et al., 2010). However, limited in time coverage, these studies remained with a static

perspective. Zodiatis et al. (2010) probably presents the most advanced vision from this approach with a hint of interannual variability.

The development of satellite observation and first SST enabled to identify some of these long-lived anticyclonic structures as accumulation areas for mesoscale eddies generated and detached from the coast (Millot and Taupier-Letage, 2005; Hamad

et al., 2006). Later altimetry products of Sea Surface Height (SSH) such as AVISO/CMEMS with grid resolution smaller than the deformation radius helped investigate mesoscale structures in the region. As in other parts of the global ocean, the turbulent eddies in the Mediterranean appeared to be prominent over the mean circulation (Mkhinini et al., 2014) and although not detected in instantaneous views, a constant and strong AW flux exists in the center of the eastern basin in a turbulent Middle-Mediterranean jet (MMJ) (Amitai et al., 2010). However, studies such as Amitai et al. (2010) used Sea Level Anomalies fields

(SLA) and an Eulerian approach of turbulence instead of eddy individual behavior. Eddy climatology in the Levantine basin remains unknown, and in particular even though warm-core anticyclones detaching from the Levantine coast were observed as early as the late 90's (Hamad et al., 2006), impact on water masses transport and exchanges performed by such transient eddies has not been studied yet.

Automated eddy detection and tracking on SSH fields and the derived velocity fields (Chelton et al., 2011) has been recently improved by algorithms that use both Local Normalized Angular Momentum (LNAM) and maximal tangential speed contours as tools to detect eddy centers and outer radius respectively (Mkhinini et al., 2014). The Angular Momentum Eddy Detection and tracking Algorithm (AMEDA) (Le Vu et al., 2018), used in this study, has improved further the tracking of individual eddies by accounting for successive merging and splitting incidents between eddies. In addition, it also corrects for cyclostrophic

balance of the surface velocity field, which allows a better representation of some intense eddies (Ioannou et al., 2019).

The AMEDA algorithm was used successfully in various case studies, notably in the Algerian basin by Garreau et al. (2018) and in the Arabian Sea by de Marez et al. (2019). Specifically, in the eastern Mediterranean sea Ioannou et al. (2017) conducted a climatological study of the Ierapetra eddy phenomenon over a 22 year period. However, the Ierapetra eddy, is unique in the

Levantine Basin as it is the only one with a marked seasonal signal (Amitai et al., 2010), but no other studies with this type of eddy detection tools have been performed on general eddy circulation in the Levantine basin so far.

Nevertheless, satellite analysis is incomplete, hiding potential differences and complexity in the subsurface structure. Moutin and Prieur (2012) for instance, showed three anticyclones in the Mediterranean sea with similar sea level anomalies (SLA)



have different subsurface isopycnal anomalies and extremely different heat and salt integrated anomalies. In the Levantine
basin Gertman et al. (2010) discovered smaller scale eddies detaching from the Israeli coast through SST and drifters data,
and Hayes et al. (2011) discovered a huge salt anomaly in the single Cyprus eddy despite a weak SSH signature. These
studies show the importance of in situ observations and the weakness of using only satellite data, but they were campaign-
specific instantaneous observations. Before eddy automated detection, Menna et al. (2012) conducted a statistical study of the
exchanges between these poles by adding in situ drifter velocities to SSH-derived velocities, but sampling was sparse and
without vertical information.

The large-scale deployment of autonomous drifters in the global ocean (such as the Argo or MEOP programs), as well as the
centralisation of collected data (such as CMEMS products), enables to bridge the gap in the temporal scale between satellite
and in situ data. Argo is a global array of more than 3000 floats measuring temperature and salinity down to 2000m (Argo,
2020). For instance, Laxenaire et al. (2019) captures the subsurface evolution of one single Agulhas ring over 1.5 years in
the South Atlantic ocean with the conjunction of Argo profiles and SSH data. It demonstrates that long-lived anticyclones can
transport warm water masses over very long distances isolated in their thick core.

Recently (Pessini et al., 2018) attempted to link eddies observations with in situ measurements and to compute eddy regional
statistics in the Algerian basin, but with an algorithm not taking into account merging and splitting events. (Mason et al., 2019)
also attempted a study of the vertical eddy structure with regional variation in the Alboran sea, but on assimilated models data.
Such regional characterisation of eddy systematic detection has not been attempted in the Eastern Mediterranean.

This approach of in situ observations colocalized with structures detected by altimetry can be generalized into an eddy atlas.
The DYNED atlas combines over 19 year of subsurface observations from Argo floats to identified eddies, tracked in time and
space, by the AMEDA algorithm with additional cyclostrophic correction over the whole Mediterranean sea (Ioannou et al.,
2019). The DYNED atlas is the perfect tool for the study of eddy dynamics and associated transport of water masses in their
cores, as it combines eddy detection and physical properties. It has not been exploited in the Levantine basin yet, although
Stegner et al. (2019) already demonstrated very deep subsurface eddy signatures in this area.

Using eddy contours, tracks and colocalized profiles from the DYNED atlas, extended with available XBT and CTD profiles
to compensate sparsity of observations, this study will investigate additional information from Lagrangian anticyclones tracking
and their contribution from their statistical behavior in terms of water masses exchanges in the southeastern Levantine basin,
east of 28° E and south of the Cyprus island. After introducing the datasets in Sect. 2, we begin with analysis of eddy activity
and lifetimes, to explain our focus on long-lived anticyclones in Sect. 3. In Sect. 4 we investigate the eddy trajectories and
exchanges between subregions of the eastern Levantine basin and in particular between the coast and areas of accumulation
offshore. A counting method of Lagrangian tracking is developed and discussed. Next oceanographic observations colocalized





inside eddies are used to study the vertical signature and water masses content of these eddies exchanges (Sect. 5). Possible
mechanisms at work are discussed in the last part.

## 2 Data

### 2.1 Eddy contours, centers and tracks

Data consisting of individual eddy tracks are retrieved from the DYNED Atlas for the period 2000 to 2018. A public release of
this database of surface intensified eddy tracks from 2000 to 2017 is available on the DYNED-Atlas website : https://www1.
lmd.polytechnique.fr/dyned/). Eddy tracks are obtained by applying the AMEDA algorithm on daily SSH AVISO/CMEMS
maps and derived geostrophic surface velocity fields. A cyclogeostrophic correction is applied on the latter to accurately
quantify eddy dynamical properties (Ioannou et al., 2019). The eddy contours considered in this study hereafter are those of
computed maximal velocity and the radius of maximal velocity is the radius of the circle with same surface than the maximal
speed contour. Eddy centers are computed by AMEDA as maxima of local normalized angular momentum. A tracking of eddy
merging and splitting events is also performed (Le Vu et al., 2018).

We focus on the eastern part of the Levantine basin (east of 28° E and south of Cyprus) by extracting a subset of eddies born
east of 24° E and south of 37° N, a definition lightly smaller than in Hamad et al. (2006). The DYNED atlas encompasses for the
whole Mediterranean 12929 eddies from 1 January 2000 to 31 December 2018 (7159 cyclones and 5770 anticyclones) whereas
the Levantine basin is a subset of approximately 2840 eddies (1630 cyclones and 1210 anticyclones), with an uncertainty of 5
units because of eddies crossing the boundaries.

### 2.2 Sea Surface Height (SSH)

The SSH used in the figures and to compute the mean dynamic topography is the 2000-2018 daily high-resolution $(1/8°)$
AVISO/CMEMS Absolute Dynamic Topography (ADT) delayed-time product provided by the Copernicus Marine Environment
Monitoring Service (CMEMS), under product name SEALEVEL_MED_PHY_L4_REP_OBSERVATIONS_008_051.

### 2.3 In situ oceanographic observations

About 34406 Argo profiles are available from the DYNED atlas in the whole Mediterranean sea over this 19-year period.
Nevertheless, due to sparser campaigns only 9384 were available in the Levantine basin. In situ oceanographic observations
are completed with 2311 CTD and 3860 XBT casts downloaded from the SeaDataNed portal ( https://www.seadatanet.org/
Data-Access, data in unrestricted access), and 7020 additional profiles from the CORA database, also available on the CMEMS
catalogue with reference INSITU_GLO_TS_REP_OBSERVATIONS_013_001_b. Finally, a glider of the Israel Oceanographic
and Limnological Research (IOLR) through one transect performed offshore Israel in October 2018 measured 370 profiles that
were added to the database. Each of these additional profiles is also colocalized with DYNED atlas, by using the maximum





velocity contours. The above add up to 22945 profiles from 2000 to 2018 in the Levantine basin. Each profile is colocalized to

the AMEDA observations, and then linked to an eddy when falling inside its maximal speed contour (Stegner et al., 2019). A climatological background is computed for each observation by averaging all profiles outside eddies from 2000 to 2019, at a distance smaller than 150 km from the observation and within a period of $\pm30$ days modulo 1 year i.e. at the same season.

## 2.4  Sea Surface Temperature (SST)

To confirm altimetric detections in some specific cases (see Sect. 4.6), we employ high-resolution ($1/120°$) merged-multisensor

SST data representative of nighttime values, provided by the CMEMS under product name SST_MED_SST_L3S_NRT_OBSERVATIONS_010_012.

## 2.5  Bathymetry

Bathymetric data used in the figures are GEBCO data with a resolution of 400m (GEBCO, 2020).

## 3  Eddy activity in the Levantine Basin

The variety of interpretations in the previous literature introduced above already highlighted the eastern Levantine basin as hosting an important eddying activity. Figure 1 presents the Mean Dynamic Topography (MDT) over the 19 years of the study in the Levantine basin with the toponyms to which we refer. The numerous closed lines of iso-MDT (hills and depressions) also show the presence of mean eddying structures. On these map, well-known features of the Levantine basin circulation can be noticed: The Eratosthenes seamount, whose summit is about 700m deep, seems to display a mean anticyclonic circulation with a

closed contour of higher MDT, at approximately 33.7 ° N ; 32.7 ° E, at the position of the formerly named "Shikmona" structure. The anticyclonic structure above the Herodotus trench can similarly be identified as the mean position of the "Mersa-Matruh" eddy. Both of these structures were previously described in the literature (Hamad et al., 2006). This MDT map also shows some variations from previous studies. In particular the "Cyprus eddy" is shifted westwards from its former reported location closer to the Levantine coast, at approximately 33.5 ° N ; 33.5 ° E (Amitai et al., 2010), having a more pronounced presence above the

Erathostenes seamount. Although there have been improvements in the SSH products since Amitai et al. (2010) (Taburet et al., 2019), this westward trend seems to be a physical displacement of the "Cyprus Eddy" inside the Shikmona structure, already reported by Zodiatis et al. (2010), as it was repeatedly surveyed in its "eastward position" in the 90's (Brenner, 1993) but more recently often above the Erathostenes seamount (Hayes et al., 2011; Moutin and Prieur, 2012).

Asymmetry between cyclones and anticyclones are enhanced in the Mediterranean sea due to strong non-linearities, an effect even more pronounced in the Levantine basin (Mkhinini et al., 2014). Figure 2 shows a comparison of the normalized cumulative eddy lifetime i.e. the probability for an eddy to live longer than a given time period. In the Mediterranean Sea, the lifetime for cyclones is on average far shorter than for anticyclones. This disequilibrium is even more pronnounced in the Levantine basin : the cyclones lifetime distribution is very similar than in the rest of the Mediterranean, whereas anticyclones

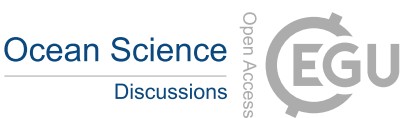

clearly tend to live longer. As an example, in absolute units of detected eddies, 105 anticyclones (out of 5770) compared to
70 cyclones (out of 7159) are found to live longer than 400 days in the whole Mediterranean, whereas in the Levantine basin
39 anticyclones (out of 1210) compared to 17 cyclones (out of 1630) live longer than 400 days. These statistics suggest the
existence of mechanisms prolonging anticyclones lifetime in the Levantine basin. Hence, this study focuses particularly on
anticyclones, as they are able to transport water masses in their thick core and as they live longer they can capture these water

masses longer. The intense Ierapetra anticyclones have been shown to live often more than 3 years (Ioannou et al., 2017), but
as no more than one of these eddies is formed each year they can not explain alone this trend of longer anticyclone lifetimes.
Moreover they rarely drift far eastwards to our region of interest in the southeastern Levantine Basin.

## 4   Eddy Lagrangian approach

As opposed to previous studies that either considered eddy activity from "building-block" structures (Robinson et al., 1991) or

Eddy Kinetic Energy (EKE) fields derived from SLA compared to a MDT (Amitai et al., 2010), we follow here eddies as daily,
individual detections gathered in tracks

### 4.1   Eddy exchange framework

In order to detect and quantify exchanges between the different subregions of the eastern Levantine basin, we define here a
framework to count eddy exchanges imported to and exported from a studied region, whose successive steps are shown in Fig.

3. The Eratosthenes seamount region, studied as an example in the above figure, is bordered with a green line (coordinates in
Table A1). First of all, we consider as *order zero* the eddies converging directly to the region (trajectories as blue lines in Fig.
3a), defined as eddies dying or spending more than half of their lifetime inside the perimeter of the region. Next, we consider as
*order 1* eddies those which merged with an order 0 eddy (Fig. 3 b, cyan lines). Recursively, we consider as *order 2* the eddies
which merging with an order 1 eddy (not shown in Fig. 3, as only 3 were found for this region). Hereafter we will discard from

the discussion *orders* higher than 2, as their quantity was found to be negligible (see legends in Fig. 6).

*Order 0-1-2* eddies are thus constituting in this framework an *importing* flux of eddies to the region. On the other hand an
*exporting* flux of eddies, is moving some eddies outside of the region. We distinguish two categories of *exporting* eddies: (1)
if two eddies undergo a splitting event and one of the split eddies spends more than half of its lifetime outside the region, it is

considered as an *exporting split* eddy; (2) if an *order 0* eddy dies while merging with external eddies which themselves drift
away from the region it is considered as an *exporting merging* eddy. Exporting split (yellow dashed lines) and merging (red
dashed lines) eddies are shown in the example in Fig. 3c ; however none of the latter case were detected for this particular
region.

It should be noted that in this framework, the *order 0* label prevails on other labels defined : if an eddy meets criteria for
*order 0* and *order 1*, it is labelled as *order 0*. Additionally *exporting (split and merging)* eddies are labelled as such only if they





are not already labelled as an *importing* eddy. It should also be noticed that no distinction is made in the *order 0* label between the ones born in the considered region and the ones born outside. Lastly, a label is relative to a region : an eddy spending more than half of its lifetime in a region but drifting and dying in another is labelled as *importing order 0* for both regions (see

discussion about Table A2 in Sect. 4.4).

In this framework and in the figures hereafter, *importing* eddies are plotted at their appearance location as black dots with size proportional to the eddy lifetime to show the origin of the water masses. *Exporting split and merging* eddies are plotted at their disappearance location as squares, also lifetime-scaled. Trajectories are smoothed through a (10 km x 10 km) window.

Color chart is summarized in Fig. 3d and used throughout this study.

## 4.2   Eratosthenes seamount attractor and attraction basin

In the particular case of the Eratosthenes seamount region, detailed in Fig. 3, anticyclones drift towards the seamount from as far as 300 kilometers away. Instead of freely drifting westwards as expected by the $\beta$-drift (Chelton et al., 2011), anticyclones

meander a lot over the high bathymetry, as shown by the density of blue tracks, seemingly trapped by the seamount. On the over hand only very few *exporting split* eddies escape and not a single *exporting merging* eddy is detected ; these split eddies are even more short-lived, as shown by the small size of the disappearance square. When adding *order 0* (blue lines) and *order 1* (cyan lines) anticyclones, this region clearly appears a particular case of anticyclonic convergence, which together with the MDT information (Fig. 1), reveals that the Eratosthenes region is not only hosting on average an anticyclonic structure but it

also consists of an anticyclone attractor. Here is then a definition of eddy convergence we can apply : eddies converge towards a given region - this region being called an *attractor* hereafter - when the importing eddies flux is significantly larger than the number of eddies born in this region, and that the exporting eddies flux is a lot smaller.

Moreover this observed convergence towards the Eratosthenes region seems to be geographically bounded, as it attracts

anticyclones from all over a region spanning from the Levantine and Egyptian coasts on the east and the south, the Cyprus island on the north and trajectories seem to stop beyond a westward limit. This area then draws an attraction basin converging to the Eratosthenes seamount and swapping a large part of the southeastern Levantine basin. We drew its border as a dashed black line on Fig. 3, defined as a straight line ranging from the Egyptian city of Alexandria (31.0° N ; 29.6 ° E) to the Cypriot city of Paphos (34.8 ° N ; 30.4 ° E) with a break angle at 33.2 ° N ; 30.4 ° E, and a second line closing it between the Greek

Cape on Cyprus island (35.0 ° N ; 34.1 ° E) to the Lebanese city of Tripoli (34.4 ° N ; 35.8 ° E). The *order 0* anticyclones form then a particular population of anticyclones meandering at this preferred position over the seamount and merging with several other anticyclones (as shown by the additional cyan trajectories), these eddies will be hereafter called *Eratosthenes attractors*.





### 4.3 Definition of anticyclonic and cyclonic regions

To study further the exchanges between structures in southeastern part of the Levantine basin, seven regions of interest are

considered - in addition to the Eratosthenes region previously introduced - and shown in the MDT of Fig. 4. The eight distinct regions are presented with solid line boxes and the attraction basin of the Eratosthenes region as a dashed line. These regions are defined as close as possible to non-overlapping rectangular shapes, tiling as much as possible the attraction basin ( $\approx 78\%$ is covered) and following when possible MDT contours. Their surfaces are similar and indicated in Table A1 of the appendix, as well as their exact coordinates. Each box is defined to correspond to a structure visible in the MDT, from which an average

cyclonic or anticyclonic activity can be inferred. Hereafter we refer to these regions as *anticyclonic (AC) / cyclonic (CY) regions* : "Beyrut" (AC), "Haïfa" (CY), "Tel-Aviv" (AC), "Port-Saïd" (CY), "Herodotus" (AC), "Nile" (AC) and "Eratosthenes" (AC). Although not in our focus region, a comparison is also done in Sect. 4.7 about the "Mersa-Matruh" (AC) region.

Figure 5 presents eddy occurrence and drift in the Levantine basin for anticyclones and cyclones, with the designated regions

of Fig. 4. Time occurrence percentage, shown in colors, is computed as the time spent inside the maximal speed contour of a detected eddy, whereas the drift, shown with arrows, is the mean speed of eddy centers passing through the pixel. Pixel size is (1/8°x 1/8 °). A Gaussian smoothing is performed (kernel size 5x5 pixels), and data from pixels crossed by less than 5 eddy centers are discarded. This picture can be seen as an eddy Lagrangian approach equivalent to the MDT shown in figures 1 and 4, adding new information. Firstly, the spatial distribution of cyclonic and anticyclonic eddies is extremely non-homogeneous

: almost no anticyclones are present in the northern part of the Levantine basin. The prevalence of the Mersa-Matruh and Eratosthenes seamount structures as persistent anticyclones is confirmed with coherent spots hosting anticyclones more than 50% of the 2000-2018 time period. Cyclones are also present in the attraction basin previously defined using anticyclones trajectories (Sect. 4.2), in particular in the "Haïfa" region. At last all *anticyclonic* (respectively *cyclonic*) regions do have on average a higher presence of anticyclones (cyclones), confirming through an eddy Lagrangian approach the definition of

anticyclonic and cyclonic regions.

Additionally comparison can be made in Fig. 5 with other well-know eddying feature such as the Ierapetra eddies. Indeed its formation site southeast of Crete can be spotted with also a strong southward drift, coherent with its observation offshore(Ioannou et al., 2017). However as Ierapetra eddies do not have a fixed stable spot after their formation, they forme a less concentrated

occurence spot in Fig. 5a which would instead reveal stable persistent structures.

### 4.4 Anticyclones Exchanges between different regions of the southeastern Levantine basin

From the average eddy activity detailed in Fig. 5 and the regions defined in Fig. 4, we can now study the exchanges between these structures. The exchange framework detailed in Sect. 4.1 is applied in the same way in Fig. 6 to the 6 other regions of the attraction basin previously defined (Beyrut, Haïfa, Tel-Aviv, Port-Saïd, Herodotus, Nile), still for anticyclones. The color code

is the same as in Fig. 3 and reminded in panel h. For each panels of Fig. 6 all other regions are shown, with green borders for



AC regions and red borders for CY ones. The region to which each panel title refers is outlined with a thicker line. Panel a is thus the simple superposition of Fig. 3a-c.

First and foremost, Fig. 6 is used to validate the coherence of the average anticyclonic or cyclonic activity characterization of each region performed through MDT in Fig. 4. It is indeed noticeable that for all AC regions (Fig. 6a-b, 6d, 6f-g), anticyclones trajectories form a rather concentrated bulk at the center of the region, whereas CY regions (Fig. 6c and 6e) have only a very sparse and random anticyclone occurrence. This confirms that only AC regions host a preference spot for anticyclones, whereas CY regions tend to be avoided by anticyclones, confirming the perspective of Fig. 5a, discussed above.

Even though each AC region hosts an average anticyclonic activity, there are important differences between them. In each region detailed hereafter the anticyclones behavior is considered characterized mostly by its *order 0* eddies. The case of the Eratosthenes seamount discussed in Sect. 4.2 revealed an attraction scheme. In the Beyrut region, anticyclones meander a lot but have few interactions (only 2 mergings and 2 splittings during 19 years) and eddies have a moderate lifetime there (126 days). In the Tel-Aviv region, anticyclones are more short-lived (97 days on average), spend less time in the region with few meanders and 3 merging exports were recorded with long-lived Eratosthenes anticyclones ; this allows us to characterize the Tel-Aviv region rather as a formation region exporting anticyclones to the Eratosthenes attractor. Characterizing the Herodotus region is more difficult, as some eddies stem from the Nile region and some others eventually merge with Eratosthenes region long-lived anticyclones ; this regions seems to act as an anticyclone formation region (42 anticyclones born there) but also transitory to the Eratosthenes one, with which it interacts a lot with 8 merging exports. On the contrary, the Nile region is a strong and preferred anticyclonic spot but eddies there interact very little with the other regions ; the Nile region acts rather as a termination region for eddies originating from the west following the Libyo-Egyptian coast. It appears that definition of a convergence region discussed in Sect. 4.2 apply only for the Eratosthenes region in Fig. 6, revealing a specificity in the southeastern Levantine basin.

Nonetheless, some anticyclones are formed inside CY regions, notably in the "Haïfa" region. In this region anticyclones are often short-lived (average lifetime 94 days) and preferably ends in the Eratosthenes attractor, with 6 mergings towards this region recorded ; the role of the Haïfa region is similar to the Tel-Aviv region acting as a region of anticyclone generation, being often exported to the Eratosthenes region. The case of the Port-Saïd region is more ambiguous, no clear pattern being visible. Statistics in appendix (Table A2) seems to show it attracts some anticyclones, but this can be due to the fact some long-lived eddies of the Eratosthenes region do sometimes venture approximately hundred kilometers farther south. This ambiguity also shows limits of this Lagrangian approach, as it is sensitive to singular event for eddies living sometimes longer than 3 years.

In addition to the individual region analysis in Fig. 6, Table A2 in appendix details all anticyclone exchanges. Regional characterisation above can similarly be studied by looking at the ratio of total importing anticyclones (order 0-1-2) over the number of anticyclones born in each region. For most AC regions this ratio is slightly above 1, showing that they might





be dynamically favorable for anticyclone generation but not attraction of new ones. However, in the Eratosthenes region the number of importing eddies is 3 times higher than the number of eddies born in the region. On the contrary the Haïfa region has a ratio of 0.79, showing that anticyclones tend to avoid this region even if they are born there, consistent with its characterization as a region of cyclonic activity. This analysis however does not stand for the Port-Saïd region, likely for the
same reason discussed above.

As a comparison, Fig. A1 in the appendix shows the same analysis as Fig. 6b for the Haïfa region but studying cyclone exchanges. Patterns for cyclones are less clear than for anticyclones, likely because cyclones live shorter and are thus more numerous (see Sect. 3). However it confirmed the Haïfa region as a stable spot for cyclones but as this region is not impermeable
to cyclones exchanges, considering it as a cyclonic attractor is less consistent.

### 4.5   Eratosthenes attractor characteristics

As previously introduced in Sect. 4.2, the Eratosthenes region acts as a regional attractor for anticyclones, which converge there from throughout the southeastern Levantine basin. Section 4.4 showed that this attraction is a specificity in the region. This attraction capacity and characteristics are discussed further here.


Figure 7a presents the individual anticyclones occupying the Eratosthenes region (*order 0 importing* anticyclones), with the color indicating when the center is inside or outside. An anticyclone is always present in this region, with relay between long-lived eddies : at almost any time period studied, one anticyclone - rarely more - occupies the place. Additionally, each time an anticyclone gets out of the region, it is quickly replaced by another. Nevertheless, it can also be noted that positions
of the long-lived anticyclone are not fixed, as they often meander out of the region. For example eddy #950 meanders almost a year further south and further west before coming back to the Eratosthenes region ; eddy #5906 drifted eastwards before dying in the Port-Saïd region several months later.

Finally, anticyclones of the Eratosthenes region are very differentiated from the other neighboring regions. Figure 7b
classifies anticyclones in terms of mean maximal speed radius as a function of lifetime ; each dot is an anticyclone labelled as *order 0* for the 7 regions studied in Fig. 6, with red color highlighting Eratosthenes anticyclones. The background is a density probability plot for all anticyclones in the Levantine basin - except Eratosthenes anticyclones to enhance comparison - . The scatter plot presents an overall spread, however long-lived Eratosthenes anticyclones form a clear cluster of eddies living longer than a year with a radius greater than 40 km, outside the 90% probability contour. Not all Eratosthenes anticyclones
are encompassed in this category, because the *order 0* label also encompasses some short-lived eddies quickly merging, which therefore unlikely have unusual characterisics. However Eratosthenes anticyclones are the only eddies in the southeastern part of the Levantine basin that present such dynamical characteristics : apart from 2 outliers, only them exceed a 40 km radius. This suggests the existence of differentiation mechanisms acting on the eddy lifetime and radius.


### 4.6  Anticyclones exchanges in the southeastern Levantine basin

All anticyclone exchanges in the southeastern Levantine basin are summarized in the Fig. 8 (Counts shown in Table A2). Exchange here encompass all *importing* anticyclones, the sum of *order 0-1-2*. Even though positions are not intended to be exact, each region studied is represented by a circle, with a blue color for AC region, and a red color for CY ones. The numbers inside circles indicate the number of attracted anticyclones (second to last row in Table A2), that is the total of *importing* eddies minus the ones that were already born in this region. White inserts indicate for each region the initial number of anticyclones

(third column in Table A2) and arrows indicate eddy exchanges, with thickness proportional to the number of eddies exchanged. Attraction capacity of the Eratosthenes region was shown with individual trajectories in Fig. 3 and it is also clear here with eddies statistics.

In the previous literature, the complex structure south of Cyprus is identified as an area of anticyclonic accumulation, called

"Shikmonah". Hamad et al. (2006) already showed with only three years of SST data that along-shore current instabilities create eddies drifting offshore (see their Fig. 16). With the hindsight of 19 years of eddy tracking, we can now quantify what was previously inferred through a limited time period of data. Anticyclones merging to the Eratosthenes region are actually often formed offshore close to the seamount on the west, in the region called in this study "Herodotus", hosting an intense eddy activity, or formed along the Levantine slope very likely due to current instabilities and drifting to this attractor. Eddy

detachments from the Israeli coast were already reported for instance by Gertman et al. (2010). On the contrary, anticyclones formed at the Nile mouth or along the Egyptian coast rarely drift offshore.

Another important result of this tracking is the isolation of this attraction basin from the rest of the Levantine basin : almost no anticyclones come from areas further west than the Herodotus region, and in the Eratosthenes region splitting events with eddies

escaping on the West of Cyprus occur but are rare (see Fig. 3 c). The attraction basin defined in Sect. 4.1 is actually extremely isolated for anticyclones. As shown by eddy drift arrows in Fig. 5, the attraction basin (dashed line) indeed coincides with divergence for anticyclones, but not for cyclones. However as cyclones do not have a thick core of homogeneous water, they contribute less to water mass transport. Furthermore as shown in Fig. 2, cyclones live significantly shorter than anticyclones, hence the assumption that water exchanges due to cyclones are less important.


Here is then the picture we can refine from Hamad et al. (2006) : the anticyclone above the Eratosthenes region plays the central role of a regional attractor, but it attracts only close anticyclones detached from the Levantine coast or formed in its neighbouring western area and its basin of accumulation (dashed line in Fig. 4) is consistently closed. This western impermeable border could be linked to the presence of the Middle-Mediterranean Jet (MMJ). Zodiatis et al. (2010) notably

suggested this jet could be feeding the Eratosthenes anticyclonic structure. But the jet could also act as a barrier for anticyclone, explaining the absence of anticyclone flux from the Mersa-Matruh area into the Eratosthenes region (see Fig. 6a).



Altimetric eddy detections have however important limitations stemming from the large spatio-temporal interpolation between satellite measurement tracks. This makes the resolution of altimetric maps ($1/8°$ in the Mediterranean Sea) not sufficient to
adequately detect small-scale structures and introduces uncertainty in the detection, especially as the internal deformation radius is small (Le Vu et al., 2018). Nevertheless, other sources of satellite images such as SST contain visible eddy signatures on them (Moschos et al., 2020). On such images, we can observe filament exchanges between eddies as well as eddies moving too fast to be detected through altimetry, as are eddies detaching from the Levantine coast. Gertman et al. (2010) already spotted such a detachment in August 2009 by means of in situ data of drifting buoys.


Here, we provide observational evidence through SST images of a similar event occurring on July 2016 where a detached warm-core anticyclonic ring, part of cyclone-anticyclone dipole, will rapidly merge with another anticyclonic eddy which will later merge with the Eratosthenes anticyclone. Figure 9 portrays this event through four daily SST image snapshots where the altimetric detection DYNED contours have been superimposed. An anticyclone not detected by surface altimetry with a
particularly warm surface signature can be spotted on the right-hand side of Fig. 9a. A cyclone with which it forms a dipole can be seen on the SST in its south-east, and is also detected by the altimetry somewhat more southwards. In panel 9b this anticyclone has moved rapidly towards the DYNED Anticyclone #10754, first detected on 10/02/2016, offshore Haifa. The track of DYNED Anticyclone #10754 is depicted with a blue line. The warm-core detached anticyclone will eventually merge with it, in 9c. A month later, in panel 9d, this anticyclone will eventually merge with the eddy on the Eratosthenes Seamount,
having transferred the warm waters and the momentum of the detached warm-core eddy.

### 4.7   Mersa-Matruh attractor

Eddy attraction to the other big anticyclonic structure of the region, called "Mersa-Matruh", can be studied in comparison to the Eratosthenes attractor. In Fig.10 the same anticyclone exchange framework defined in Sect. 4.1 is applied to this region. Firstly, it is noticeable that this structure clearly also acts as an attractor, the total number of *importing* eddy (sum of *order*
*0-1-2*) being a lot higher than *order 0* alone and *exporting* eddies. In a similar fashion with the Eratosthenes region, which acts as a stranding place for anticyclones detached from the Levantine coast, lots of anticyclones detached from the Libyo-Egyptian coast (likely as instability of the Libyo-Egyptian current) end into the Mersa-Matruh, often trough one merging or more. In particular a hotspot of anticyclone formation takes place in the Mersa-Matruh gulf approximately at ($31.5°$N , $27.5°$E) from which 5 drifted towards the Mersa-Matruh structure. No importing anticyclones come from the north, as expected given that
no anticyclones occur in the Rhodes gyre (see Fig.5a).

However in contrast to the Eratosthenes attraction basin being isolated with very few anticyclone exchanges outside, the Mersa-Matruh anticyclone has no clear western boundaries. For instance some merging export trajectories in red are present southwestwards, pointing out that some eddies escaped from this structure. On the contrary, one Ierapetra eddy did end through
successive mergings into the Mersa-Matruh area. This individual event seems to be isolated, as shown in Ioannou et al. (2017), Ierapetra eddies actually tend to go westwards riding up the Libyo-Egyptian current if they move away, in a similar way to



the eddy behaviour described by Sutyrin et al. (2009). More generally, the importance of *higher order* anticyclones merging in successive steps to Mersah-Matruh anticyclone suggests that convergence is less straightforward and clear than for the Eratosthenes seamount, likely because topographic constraints are less present.


# 5 Vertical structure

## 5.1 In situ data colocalisation

A big improvement of the DYNED atlas was to colocalize oceanographic measurements to eddy observations, when in situ profiles fall inside the altimetric detection contours, as explained in Sect. 2.3. A characterization of the average physical
properties of the eddies can be achieved by building a climatological background through an average of all profiles outside eddies (Stegner et al., 2019), the heat, salt and density anomalies associated with each eddy can be computed, provided that the eddy was surveyed by an oceanographic instrument.

For very persistent long-lived eddies, like the Eratosthenes anticyclone attractor, this allows to observe changes in the vertical
structure and the evolution of the associated anomalies. Figure 11 shows for different years during the 2000-2019 period the annual averaged vertical profiles of the Eratosthenes attractors, over all available profiles within each year, and closer than 30 km to the eddy center. A histogram below indicates for each year how many profiles are available. This number varies a lot due to inconstant frequency of oceanographic surveys, with years 2008 to 2011 being over-represented because of extensive glider sections (Hayes et al., 2011) and several Argo floats being trapped for a very long time inside the anticyclone. Increasing the in
situ DYNED database, initially only based on Argo floats allows to sample more accurately the eddies and especially to have a better timeseries, CORA and SeaDataNet being a lot more provided in the 2000-2010 decade.

In the annual averaged vertical profiles in Fig.11, the Eratosthenes anticyclones can be characterized by very deep anomaly, both in salt and temperature. For every year the depth of maximal density anomaly is below 200 m and reaches some years
450 m. Magnitude of the anomalies can also be extremely marked, higher than 2.5°C and 0.45 PSU in 2010. However, if annually averaged temperature anomalies all reach 1 °C at 200 m or below, Fig.11 especially shows that there is a strong interannual variability in the vertical structures of these anticyclones. 2010 then appear as an extreme event, with the formation of a double-core structure visible on the density profile (panel 12c). This event was surveyed by gliders and described by Hayes et al. (2011), but now with longer timeseries it can be seen that anticyclone anomalies were extreme compared to the 19 years
mean vertical structure.

Moutin and Prieur (2012) also compared in the BOUM campaign in 2008 the vertical structure of an Eratosthenes anticyclone and 2 others ones in the Mediterranean sea : a detached Algerian eddy and an anticyclone in the central Ionian sea. Their comparison showed that the Eratosthenes anticyclone had the deepest potential density maximal anomaly - 380 m in Moutin





and Prieur (2012) - and that integrated anomalies were very warm and salty waters. However as shown in Fig.11b-c the magnitude and depth of such anomalies can greatly varies from one year to another. Depth of maximal density anomaly often below 300 m are nonetheless an almost unique specificity in the whole Mediterranean sea (Stegner et al., 2019).

## 5.2 Anticyclones comparison

The vertical structure of Eratosthenes anticyclones described above should be compared with physical properties of neighbouring
eddies. Figure 12 presents the comparison between a section representative of the Eratosthenes anticyclone, using data from the BOUM campaign in June 2008 (Moutin and Prieur, 2012), and an anticyclone section in the Tel-Aviv region performed in October 2018 by a glider from IOLR. Next to each section is an ADT map representative of the SSH activity at that time (panels b and d). The glider section lasts for 10 days, 20th of October 2018 being the median date and a magenta line indicates the glider track, on which the position on the 20th of October is shown with a magenta circle. On the ADT map the daily
altimetric eddy contours are plotted, with cyclones in red and anticyclones in blue, and a dot with size proportional to the vortex Rossby number indicates the center. The upper panel of each section (panels 12a and 12c) marks with a blue line the part of the cross-section which is inside the maximal speed anticyclone contour. Thin black lines in the vertical sections are the absolute temperature isotherms, whereas the color indicates the temperature anomaly relative to the climatological background ; isotherms interval and colorbar range are the same in both sections for comparison purposes. An important difference is that
panel 12a is an interpolation between the CTD measurements (indicated by black crosses) whereas panel 12b shows a glider track stacking of which each pixel corresponds to a measurement.

Although a 10-years period separates the two sections, it should first be underlined that the local eddy activity is similar in both events, and very close to the mean circulation deducted from Fig.1 : Eratosthenes, Herodotus, and Tel-Aviv regions are all
occupied by anticyclones (also the Beyrut one but only in 2008), whereas Haïfa region is occupied by a cyclone. Furthermore as can be seen in the ADT maps, both sections crossed the anticyclones close to their respective center, allowing to assume that the maximal anomaly was adequately sampled. Extensive gliders sections were surveyed in the Eratosthenes region in 2009 and 2010, but as explained above, 2010 appears as an extreme year where comparison with neighbouring anticyclone will be biased.

The "Eratosthenes" eddy surveyed in Fig.12 (a) is a long-lived anticyclone, referenced in the DYNED database with number 4914, born in the Beyrut region before settling at the location of the Eratosthenes attractor for more than 6 months. The "Tel-Aviv" eddy measured in Fig.12c is young anticyclone, formed close to the shore in August 2018 as detected by AMEDA, at the approximate position 32.0 ° N, 34.0 ° E, and referenced in the DYNED database with number 12683. It drifted slowly offshore northeastwards before dying without merging at approximately 32.5 ° N ; 33.0 ° E at the beginning of December 2018. It is
therefore very similar to the anticyclones formed in the Tel-Aviv region and drifting offshore, sometimes achieving to merge in the Eratosthenes region, in a similar way to the trajectories shown in Fig.6c.





In both events the "Tel-Aviv" anticyclone appears to be more intense in terms of Rossby number than the "Eratosthenes" one, but the vertical structure shows that the weak Eratosthenes anticyclone actually hides a very deep and strong temperature
anomaly, reaching 2.2°C at its core ( 1.3 °C for the "Tel-Aviv" one).

## Discussion

This study described the convergence of anticyclones in the southeastern Levantine basin towards the region of the Eratosthenes seamount, but questions remain regarding the mechanisms. At first the existence of attractor structures represented by long-
lived anticyclones is observed whereas more classical westwards $\beta$-drift or current advection along the coast could be expected (Sutyrin et al., 2009). A question could be whether the long-lived structure effectively attracts other eddies or if eddies detached from the coast and merge with a central and long-lived structure. SST images and detachments observed from the coast, shown in Fig.9 in this study but previously observed in the literature by Hamad et al. (2006) and Gertman et al. (2010), suggest the second option is likely happening. Further studies in this direction are needed to outline eddy dynamics, but great improvements
are possible with eddy automatic detection at much smaller scale with the help of SST data (Moschos et al., 2020).

These attractor structures occur above the Eratosthenes seamount as studied here but also very likely in the Mersa-Matruh region. Such structures were already surveyed in the 80's (Brenner, 1993; Robinson et al., 1991; Hamad et al., 2006) and identified as area of eddy accumulation as previously explained, but here we described a very different structure, in term
of attraction behavior and marked in depth physical properties anomalies. Stationary eddies are surveyed in other places, for example in the Alboràn sea, (Mason et al., 2019), and also anticyclones with unusual characteristics and lifetime, for example Ierapetra eddies (Ioannou et al., 2017). However no other anticyclone achieve such stationarity - as shown in Fig.5a - together with strong vertical anomalies (Hayes et al., 2011; Moutin and Prieur, 2012). Topographic constraints could be working to maintain a marked anticyclone above the Eratosthenes seamount, but actually the presence of a seamount should
more disadvantage anticyclone development than favour it (Sutyrin et al., 2011). Further work on Mersa-Matruh would be needed, but nonetheless the stationarity of both Eratosthenes and Mersa-Matruh anticyclones in the Levantine basin although the first is on top of a 700 m deep seamount and the second one above the Herodotus abyssal plain below 2500 m also strengthens the idea that topography is not the leading parameter in long-lived anticyclone dynamics.

A third point raised in this study is the effective barrier for anticyclone between what we defined as a convergence area towards the Eratosthenes region and the central Levantine basin. This isolation stands out by comparison to anticyclone drift over long distance in the western Levantine basin (Ioannou et al. (2017) with examples of Ierapetra eddies). This boundary might be linked to some stationary feature like jets acting as a barrier, a stationary portion of the Middle Mediterranean Jet as proposed by Amitai et al. (2010) being a good candidate. This jet could then be partly represented in the MDT gradient between
Alexandria, the Nile fan and Cyprus, visible on Fig.4 and explain the absence of eddy leakage between the Mersa-Matruh and

Eratosthenes regions, both captured in the MDT. In older literature, Robinson et al. (1991) made the hypothesis of a northern path of Atlantic waters in the Levantine basin, letting the south-eastern part of it isolated, but still with Mersa-Matruh and Eratosthenes regions together. As discussed in Hamad et al. (2006) this path is inaccurate as the main path for Atlantic waters following the Mediterranean coast. The isolation of this south-eastern part of the Levantine basin for anticyclones presented in

this study could also indicate that Robinson's schema is not totally inaccurate for eddy drift

Last but not least given the importance of this area for intermediate waters formation, the fate of watermasses in the Eratosthenes anticyclone core and its dissipation would be of great interest for extended research. Whether or not it helps sustaining the structure, we proved in this study that eddies with different origin regularly merge with these large and deep-

cored anticyclones, and the imported water masses should go somewhere. They likely feed its subsurface anomaly, but later the final destination of these waters is unknown. Erosion from its deep anomaly due to intense shear with topographic interaction (Sutyrin et al., 2011) could give a warm and salty water flux in depth and leading to intermediate waters formation. Answering this very important question implies further simulation work, but also a lot more oceanographic data forming a consistent and continuous time series to accurately follow its evolution.

**6   Conclusions**

Eddy Lagrangian tracking in the eastern part of the Levantine basin over 19 years of data from 2000 to 2018 allowed to spot persistent anticyclone convergence over the Eratosthenes seamount from a specific attraction basin in the southeastern Levantine. The distinction of mean eddying activity regions enabled to quantify anticyclone exchanges in this part of the basin, and revealed that most anticyclones merging toward the Eratosthenes seamount were actually formed in a nearby

area westwards or more remotely southeastwards near the coast offshore Tel-Aviv. The attraction basin defined appears to be almost impermeable for anticyclone drift. Statistical analysis of the dynamical parameters of these anticyclones shows that the Eratosthenes attractors are larger (over $40km$ in radius) and live very long (more than 1 year and up to 3). In situ oceanographic profiles colocalization also revealed that contrary to most neighboring eddies they have a strong subsurface structure, with associated temperature and salinity anomalies always extremely marked and deep, depth of maximal depth

anomaly being always at or below 200 m and reaching some years 450 m, but also with pronounced interannual variability.

*Author contributions.* A. Barboni did the main analysis and wrote the manuscript ; A. Lazar supervised the study and provided the glider data ; A. Stegner supervised the study ; E. Moschos analyzed SST data and produced Fig.9. All authors contributed to finalize the manuscript.

*Competing interests.* The authors declare that they have no conflict of interest.



*Acknowledgements.* This research was supported by THE ISRAEL SCIENCE FOUNDATION (grant No. 1666/18)



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



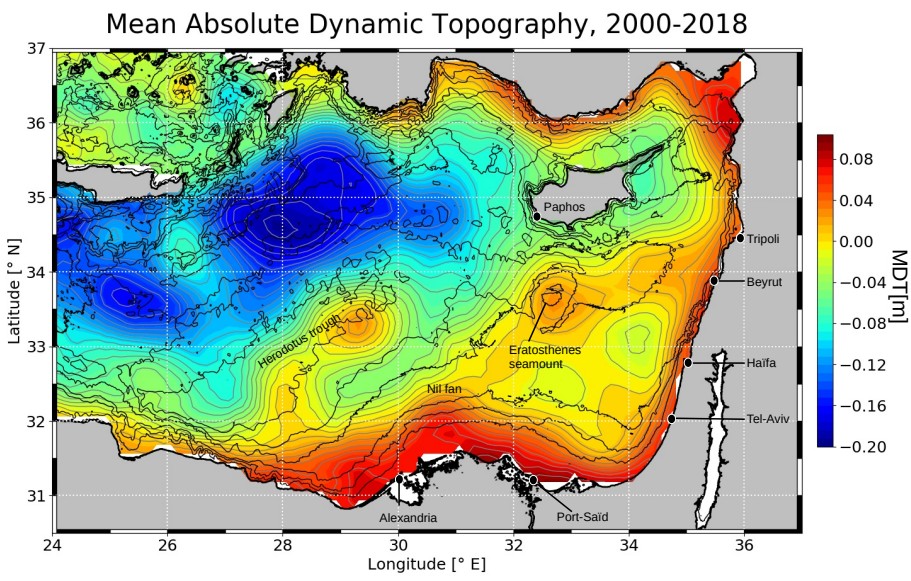

**Figure 1.** MDT map of the Levantine basin and several toponyms and city names used in this study. Thin black lines are the -100, -500, -1000, -1500m, -2000m and -2500m isobaths, thick black line is the 0m isobath.





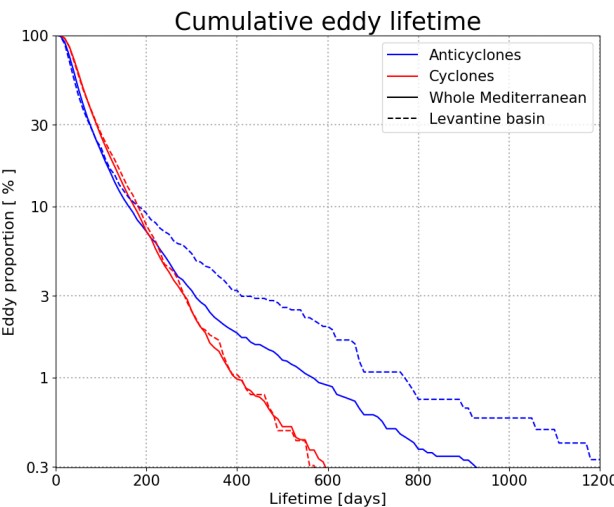

**Figure 2.** Cumulative eddy lifetime as percentage in logarithmic scale, separated for cyclones and anticyclones and for the Levantine basin and the whole Mediterranean.

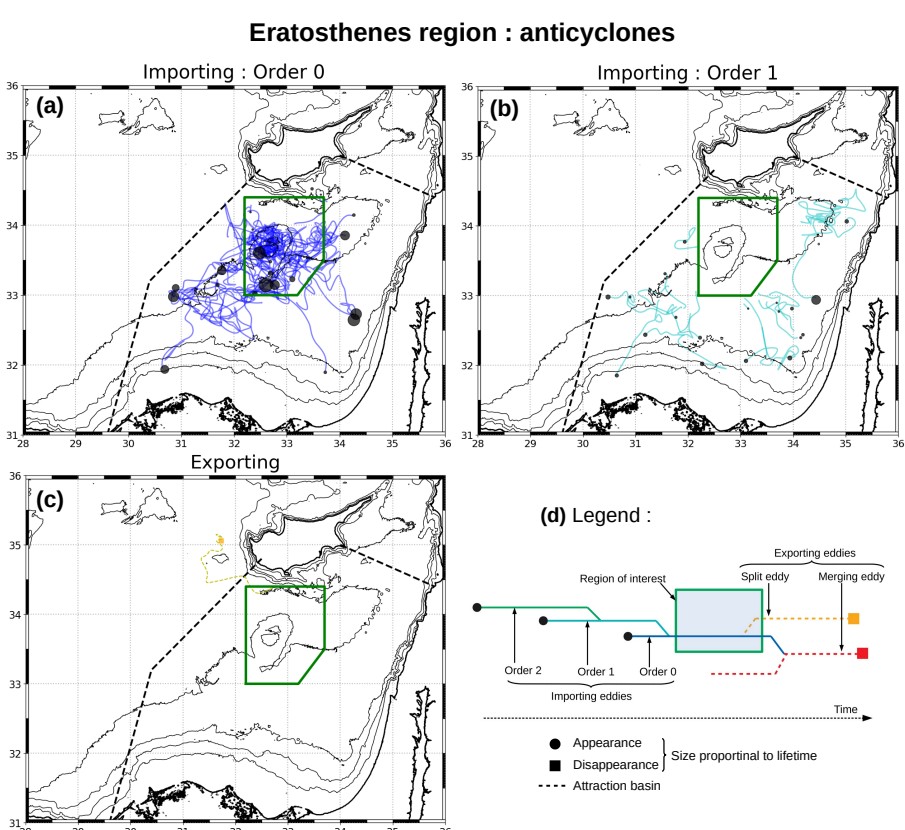

**Figure 3.** Lagrangian framework applied to anticyclones in the Eratosthenes seamount region and detailed in several steps. (a) *Order 0* : eddies converging directly ; (b) *Order 1* : eddies merging with an *order 0* eddy ; (c) *Exporting split* and *merging* eddies (no cases of the latter detected here). Locations of eddy appearance for (a) and (b) (respectively disappearance for (c)) are shown with black dots (respecively colored squares) whose size is proportional to the eddy lifetime. (d) Legend.



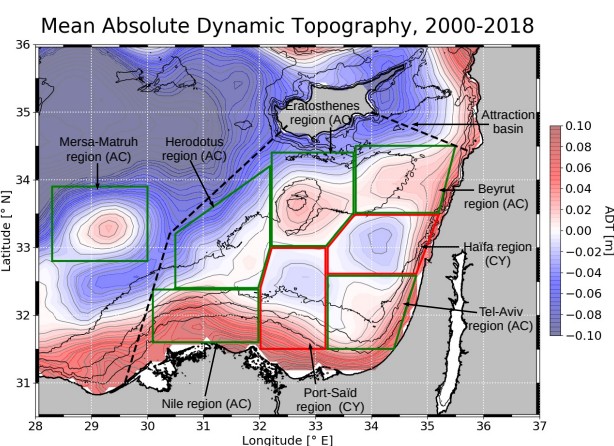

**Figure 4.** MDT map of the eastern Levantine basin labelled with borders and names of the studied regions. Anticyclonic regions have a green border, cyclonic regions a red one, boxes coordinates being indicated in appendix (Table A1). The attraction basin (dashed black line) is defined in Sect. 4.2.


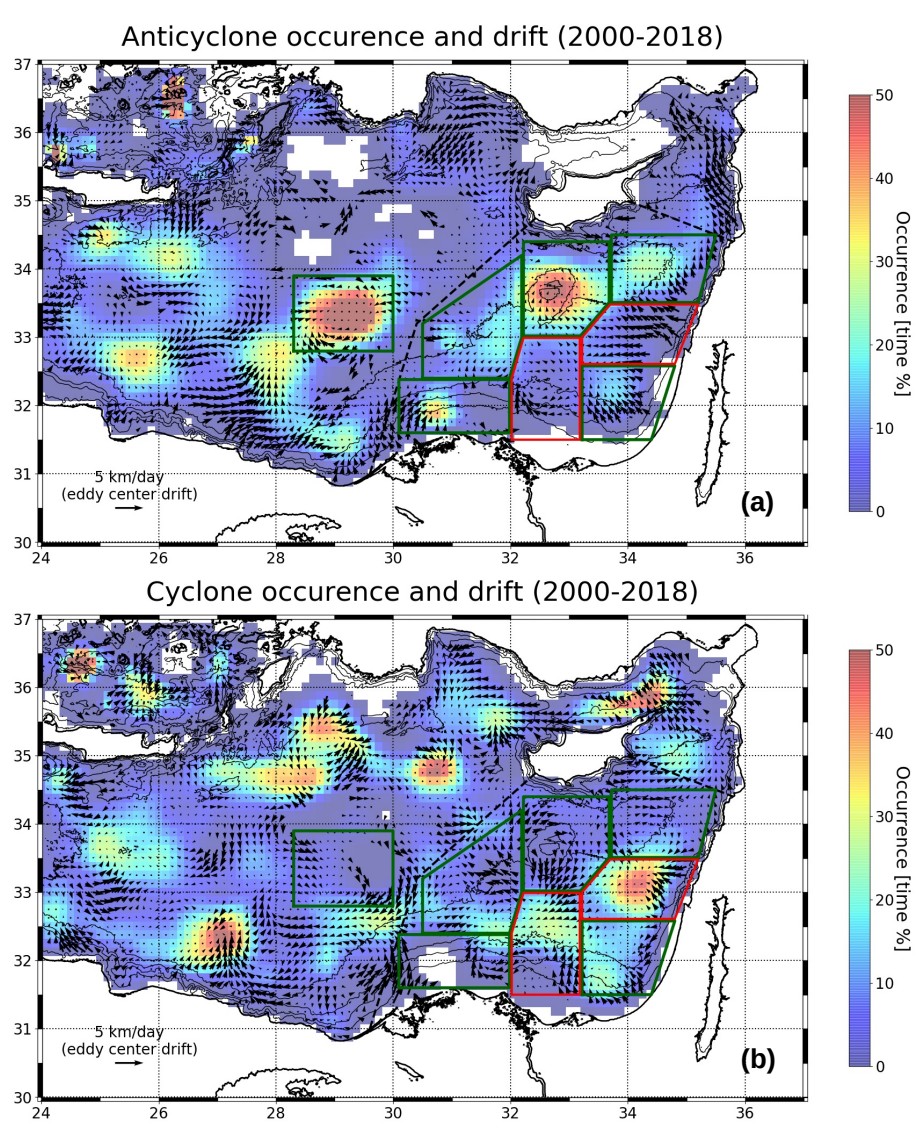

**Figure 5.** Eddy occurrence and drift in the Levantine basin for (a) anticyclone and (b) cyclones. Pixel size is (1/8°x 1/8°). Occurrence is shown as the time percentage when the pixel center spends inside maximal speed contours of eddies. Drift is the mean Lagrangian drift of the eddy centers on pixels where more than 5 eddy centers passed, with Gaussian smoothing (kernel size (5x5)).





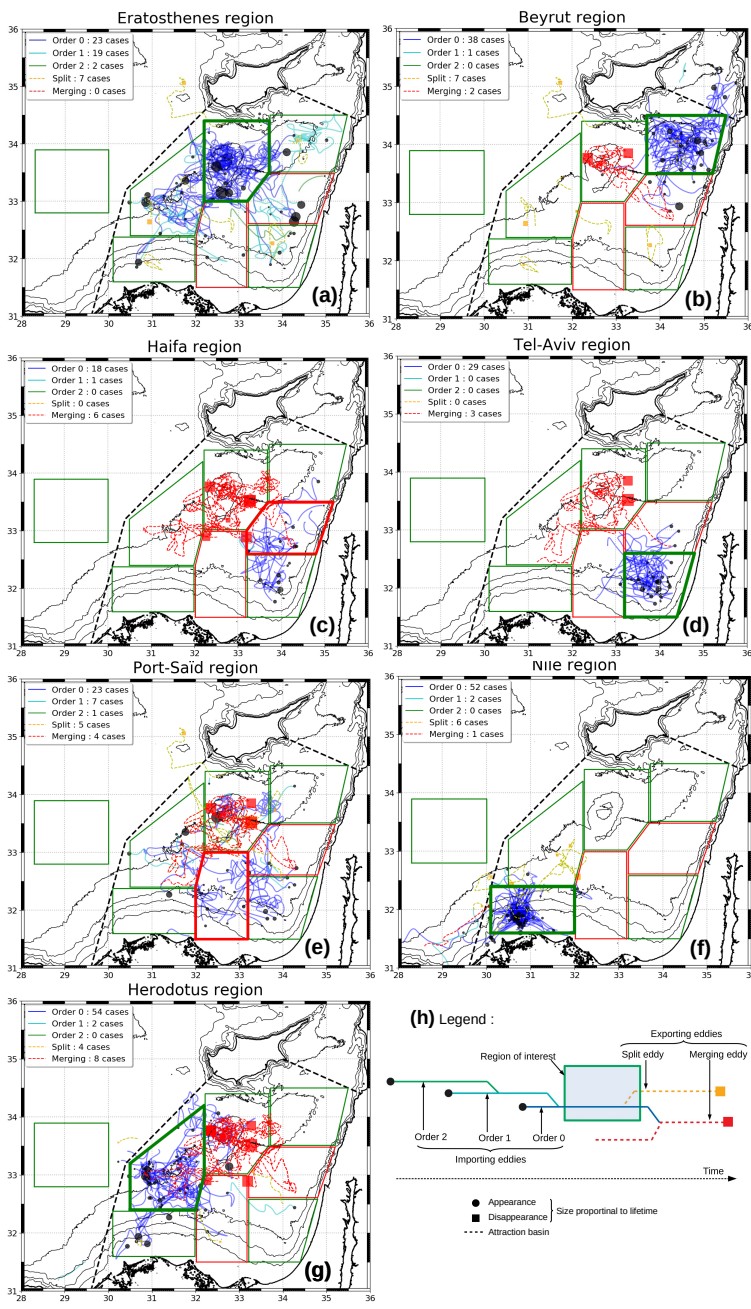

**Figure 6.** Eddy exchanges framework applied to (a) Eratosthenes, (b) Beyrut, (b) Haïfa, (d) Tel-Aviv, (e) Port-Saïd, (f) Nile and (g) Herodotus region. For each panel all other regions are shown (green borders for AC regions, red borders for CY ones), while a thicker line indicates the studied region. Color chart used is summarized in panel (h). Mersa-Matruh region is studied later in figure 10.


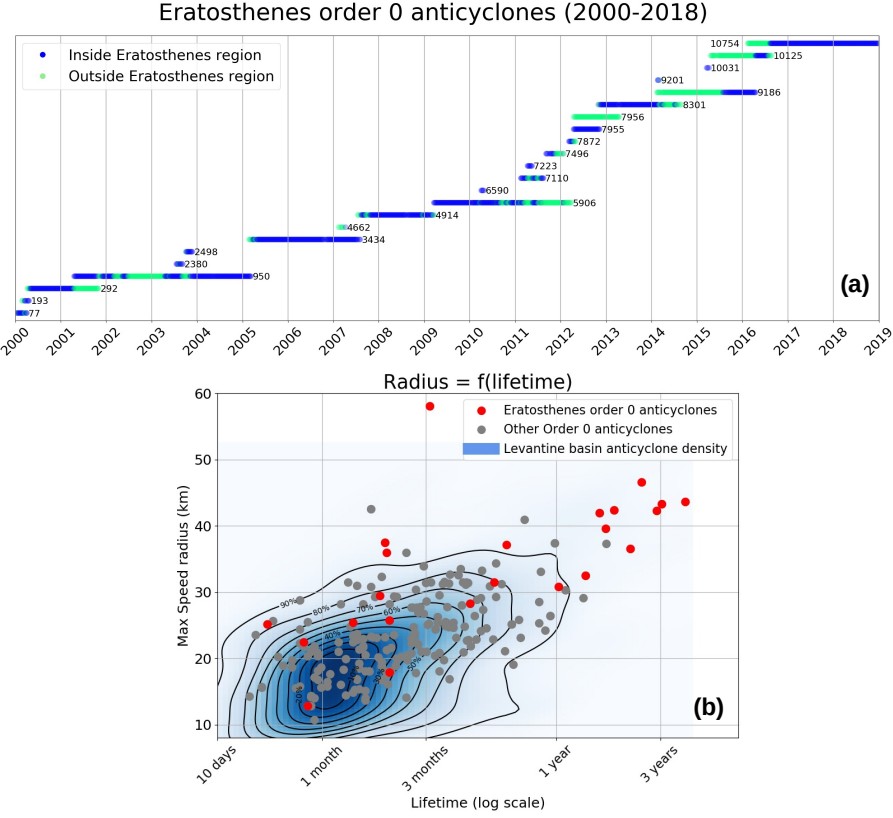

**Figure 7.** (a) : time series of *order 0* importing anticyclone to the Eratosthenes region, with eddy ID number in the DYNED database. The color marker indicates when the eddy center is within (blue) or outside (green) the region. Corresponding trajectories are figure 3a. (b) : Scatter plot of maximum speed radius as a function of lifetime, in logarithmic scale, for *order 0* importing anticyclones of the Eratosthenes region (red dots) and the other regions (gray dots) ; blue shaded background is the density function for the whole Levantine basin anticyclones - except Eratosthenes ones - with contours each 10% probability step. Different regions studied in figure 6 are detailed in supplementary figure A2.



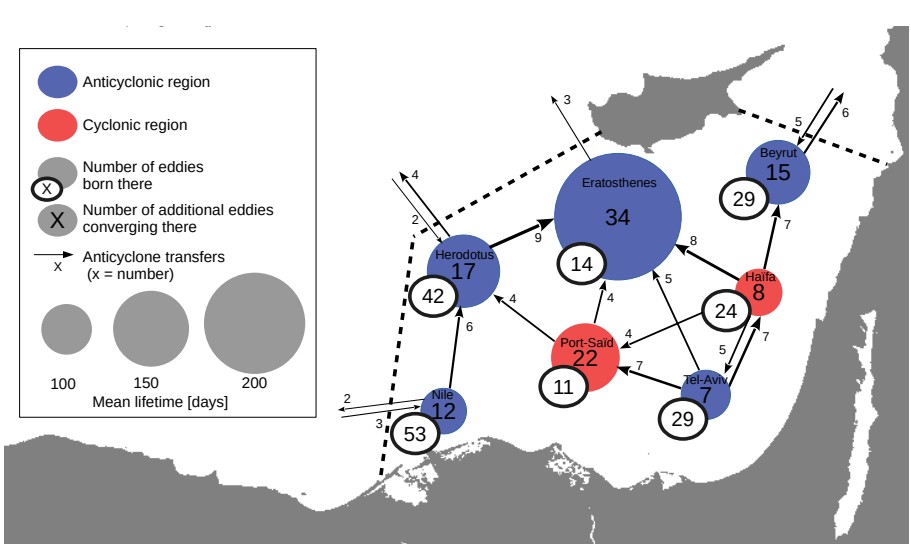

**Figure 8.** Scheme summarizing anticyclone exchanges from 2000 to 2018 within the attraction basin (dashed line). Transferts encompass *order 0-1-2* anticyclones. Within this basin, only transfers higher or equal to 4 are represented, but across the basin border all transfers are shown. Arrow thickness is proportional to eddy exchanges. Red circles represent CY regions, blue circles AC ones, with radius proportional to anticyclone mean lifetime in the region. All details are resumed in Table A2 in appendix.



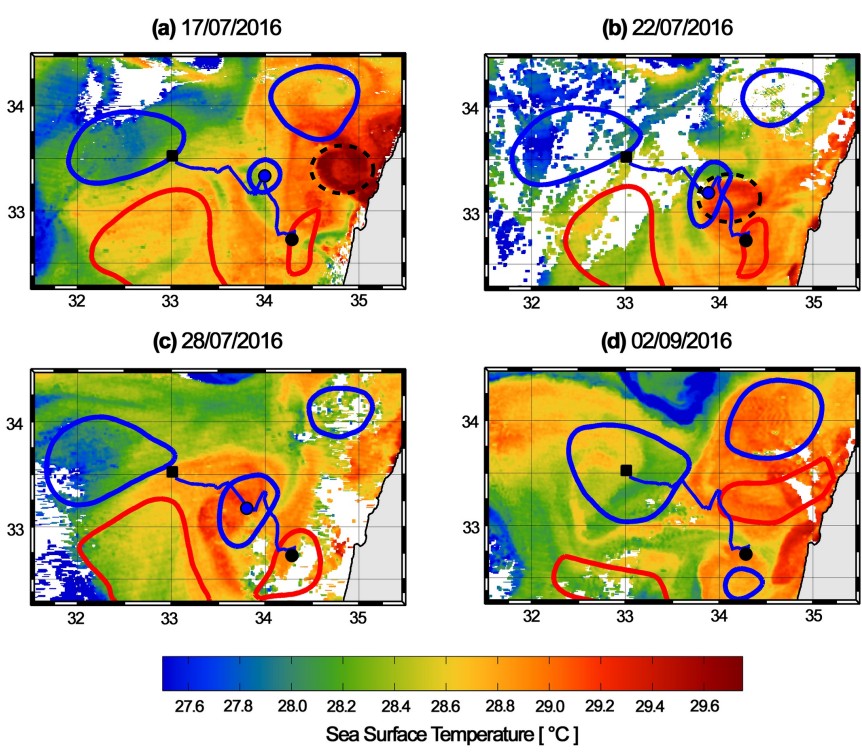

**Figure 9.** Daily snapshots of SST images showing a quickly moving warm-core anticyclone, part of a dipole, which detaches from the Levantine coast and merges with a future Eratosthenes anticyclone. Superimposed DYNED contours are blue for anticyclones and red for cyclones. The track of the studied anticyclone is in blue line, its current center with a blue circle, its initial detection on 10/02/2016 with a black circle and its last detection on 02/09/2016 with a black square. A blue line approximates the track of the warm-core anticyclone as seen in the SST sequence.





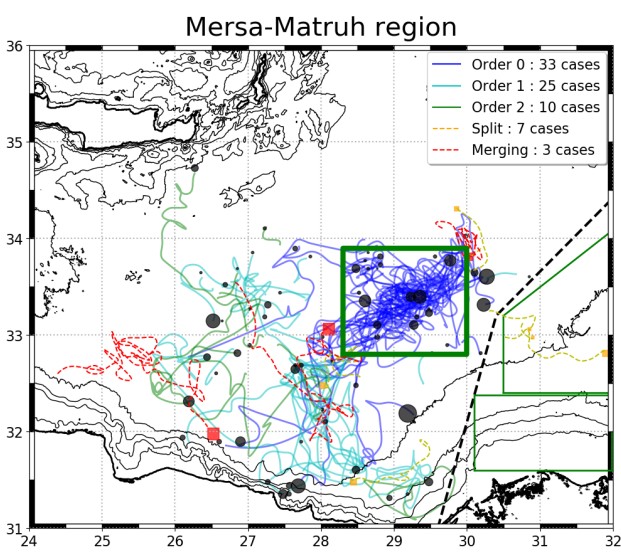

**Figure 10.** Convergence structure applied for the Mersa-Matruh region, in the same way and same color code as in figure 6.



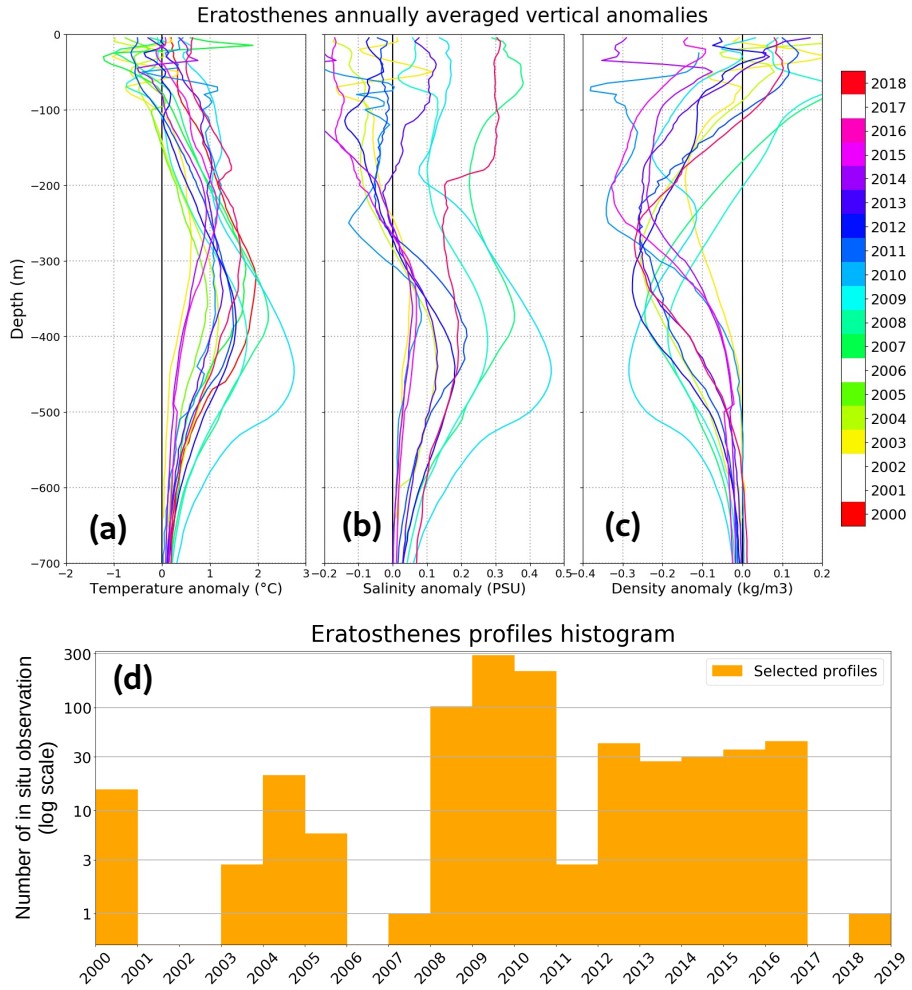

**Figure 11.** For each year, average of the vertical profiles anomalies, for (a) temperature, (b) salinity and (c) density, inside Eratosthenes *order 0* anticyclones. Profiles are selected if they are cast within 30 km from the eddy center and their histogram is shown in panel d, with a log scale. Years without any profiles are discarded from the colorbar.



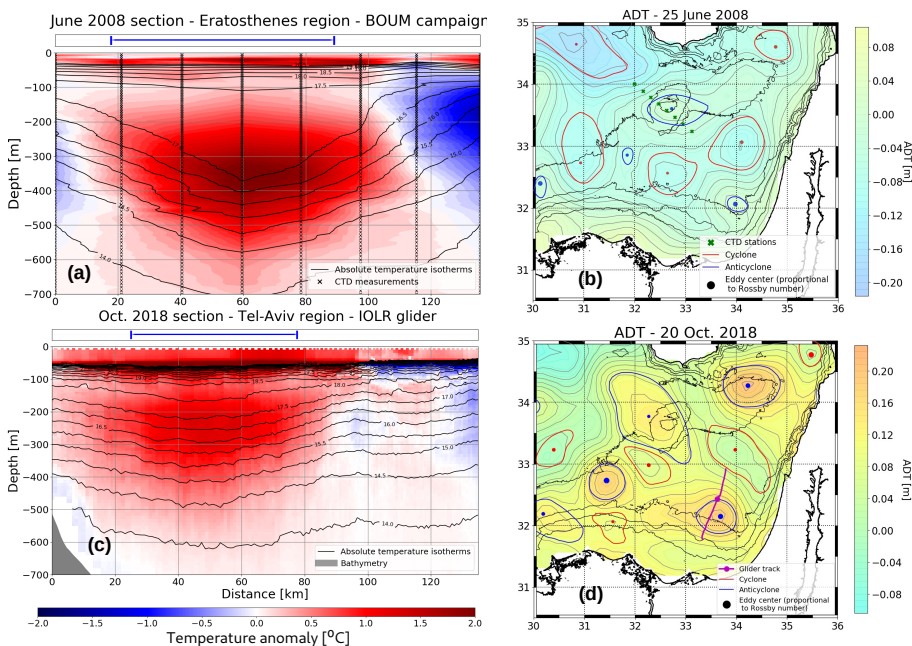

**Figure 12.** Comparison of anticyclone sections. Left panel (a) (respectively (c)) : Eratosthenes (Tel-Aviv) region anticyclone section in June 2008 (mid-October 2018) ; north on the left side ; horizontal axis (shared) is distance along the section ; black lines are absolute temperature isotherms and the temperature anomaly as color background ; blue cartridge above the panel reminds maximal speed limits. Right panel (b) (respectively (d)) : ADT map presenting the eddy neighboring eddy activity, green crosses (a magenta line) showing the CTD casts (the glider track) ; eddy contours are maximal speed contours, blue for anticyclones, red for cyclones, with a Rossby number-scaled dot for its center.

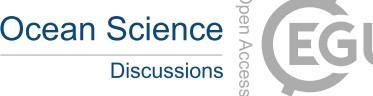

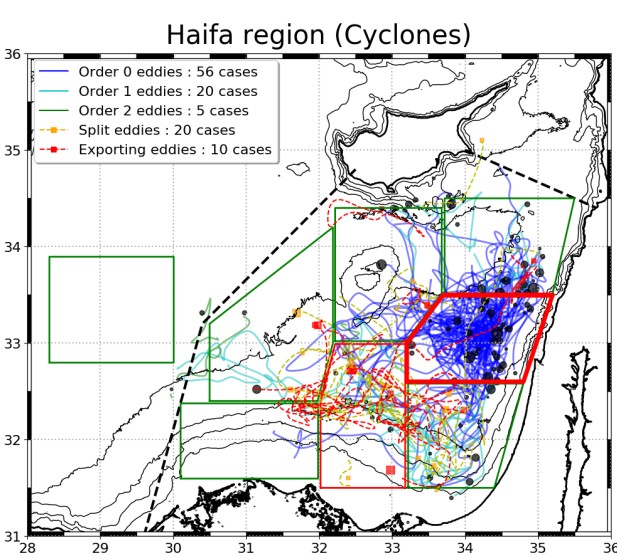

**Figure A1.** Cyclone exchanges for Haïfa region. Same as Fig.6c but with eddy exchange framework applied for cyclones. Boxes and dashed line are the same as in Fig.4.





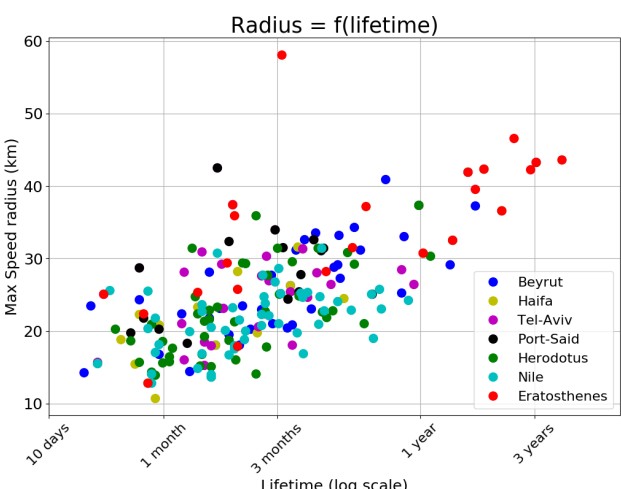

**Figure A2.** Same as Fig.7b but without density plot. Scatter plot of maximum speed radius as a function of lifetime, in logarithmic scale, for *order 0 importing* anticyclones of all region studied in Fig.6.




**Table A1.** Regions studied in the Levantine basin, with coordinates and area. Corresponding boxes are shown in Fig.4.

| **Region** | Eratosthenes | Beyrut | Haïfa | Tel-Aviv | Port-Saïd | Herodotus | Nile | Mersa-Matruh |
|---|---|---|---|---|---|---|---|---|
| **Coordinates** | 33.0 ; 32.2 | 33.5 ; 33.7 | 32.6 ; 33.2 | 31.5 ; 33.2 | 31.5 ; 32.0 | 32.4 ; 30.5 | 31.6 ; 30.1 | 32.8 ; 28.3 |
|  | 33.0 ; 33.2 | 34.5 ; 33.7 | 32.6 ; 34.8 | 32.6 ; 33.2 | 32.4 ; 32.0 | 33.2 ; 30.5 | 32.4 ; 30.1 | 32.8 ; 30.0 |
|  | 33.5 ; 33.7 | 34.5 ; 35.5 | 33.5 ; 35.2 | 32.6 ; 34.8 | 33.0 ; 32.2 | 34.2 ; 32.2 | 32.4 ; 32.0 | 33.9 ; 30.0 |
| **(°N , °E)** | 34.4 ; 33.7 | 33.5 ; 35.2 | 33.5 ; 33.7 | 31.5 ; 34.4 | 33.0 ; 33.2 | 33.0 ; 32.2 | 31.6 ; 32.0 | 33.9 ; 28.3 |
|  | 34.4 ; 32.2 |  | 33.0 ; 33.2 |  | 31.5 ; 33.2 | 32.4 ; 32.0 |  |  |
| **Area ($\times 10^3\,km^2$)** | 20.7 | 17.3 | 15.7 | 16.1 | 18.3 | 22.6 | 16.0 | 19.7 |




**Table A2.** Anticyclones exchanges between regions of the Levantine basin, illustrated by the diagram in Fig.8. The attraction basin is defined in Sect. 4.2. Row 'Attracted' shows how many anticyclones ends in a given region while not born in this region. For example in the Eratosthenes region this quantity is : $44 - 10 = 34$. Blank means the quantity is zero, while '/' means it is not defined. In columns following abbreviations are used : 'Er.' for Eratosthenes, 'B.' for Beyrut, 'TA' for Tel-Aviv, 'Her.' for Herodotus, 'Hf.' for Haifa, 'PS' for Port-Said, 'Else in' for 'elsewhere inside the basin' and 'Else out' for 'elsewhere outside the basin'.

| | | Anticyclones born | Termination region | | | | | | | | |
| --- | --- | --- | --- | --- | --- | --- | --- | --- | --- | --- | --- |
| | | | Er. | B. | TA | N. | Her. | Hf. | PS | Else in | Else out |
| AC | Eratosthenes | 14 | 10 | 1 | | | 3 | | 3 | | 3 |
| | Beyrut | 29 | 3 | 24 | | | | | 1 | | 6 |
| | Tel-Aviv | 29 | 5 | | 22 | | 1 | | 7 | | |
| | Nile | 53 | 2 | | | 42 | 6 | | 3 | 1 | 2 |
| | Herodotus | 42 | 9 | | | 1 | 39 | | 3 | | 4 |
| CY | Haifa | 24 | 8 | 7 | 5 | | | 11 | 4 | | |
| | Port-Said | 11 | 4 | | 2 | 1 | 4 | | 9 | | |
| Else | Inside the basin | 27 | 3 | 2 | | 7 | 1 | | 1 | 4 | 12 |
| | Outside | 19 | | 5 | | 3 | 2 | | | 12 | / |
| | TOTAL | 248 | 44 | 39 | 29 | 54 | 56 | 19 | 31 | 17 | 27 |
| | Attracted | / | 34 | 15 | 7 | 12 | 17 | 8 | 22 | 13 | / |
| | Mean lifetime (days) | 121 | 254 | 126 | 97 | 90 | 117 | 94 | 136 | 39 | 109 |

