# Peer review of "Lagrangian eddy tracking reveals the Eratosthenes anticyclonic attractor in the eastern Levantine basin"

_Ocean Science, 2020_

## Author Response (AR1)

**Reply to reviewer comment 1**

**General answer :**

We thank the reviewer for sharing his/her concerns about the eddy tracking methodology. Taking into account the review remarks, several changes were made to the manuscript, in particular to explain more clearly the methodology used. We seek here to clarify the reviewer's remarks with supplementary figure material.

Regarding the objectivity of the eddy detection and tracking method used for this research, we clarify in the introduction that most of the previous studies (Chelton et al (2011), Mason et al (2014)) did not take into account merging and splitting eddies. Several more recent eddy detection algorithms were developped to detect such events, using various parameters : SSH ( Matsuako et al. (2016), Cui et al. (2019) ), mixed Okubo-Weiss-SSH approach (Yi et al (2014)) or mixed velocity field-SSH (AMEDA, Le Vu et al (2018), used in this study).

However, even though some of the above-mentioned studies map the occurrences of detected eddy interactions, we are not aware of studies that have quantified nor analyzed the additional information  from merging and splitting events in terms of eddy behavior, apart from the work of Laxenaire et al (2018) for Agulhas rings. It should also be noted that these events are still not taken into account in more recent studies on Mediterranean eddies (Pessini et al. 2018, Mason et al. 2019).

As a main answer about the potential biases in the count strategy, we provide here some additional plots. In Fig. R1 an overall convergence is clear for *order 0 importing* anticyclones in the Eratosthenes region, often born far from Eratosthenes seamount and dying within the region borders or very close to it (dots and squares are scaled by the eddy lifetime). Encompassing in the *order 0* definition the eddies that do not die in the studied region but spend 50 % of their lifetime inside it then does not lead to inaccurately count transiting eddies. Indeed if leaving the region, *order 0* anticyclones do not drift far away.

However we acknowledge that this methodology might need to be adapted when studying other regions of the world ocean where eddies can drift away over thousands of kilometers (e.g. South Atlantic ocean in Laxenaire et al, 2018), which is not true in the Mediterranean sea.

With a careful look, we found one occurrence of what could be considered as a transiting eddy : anticyclone DYNED ID 10125 is born in the Nile region, drifted northwards in the Herodotus region where it spent more than 50% of its lifetime, and eventually died within Eratosthenes region. This eddy is then labeled as *order 0* for both Eratosthenes and Herodotus region, the later not being consistent. Such a low occurrence rate seems to be satisfying.

[Figure]

**Figure R1 :** Appearance and disappearance locations for all *order 0* anticyclones, for the Eratosthenes region. Even if not all of them die within the region, they disappear very close to it.

The sensitivity of this 50 % lifetime criterion is analyzed in Fig. R2 (also added in the manuscript appendix), looking at the changes in total *order 0 importing* anticyclones as a function of this threshold. "100%" means *order 0* anticyclones are strictly and only the ones dying in the studied region, "30%" means that *order 0* are the ones dying in the studied region, **plus** the ones spending 30% of their lifetime within the borders of this region. Consequently the ordinates for 50% are the *order 0* counts in this study (and are retrieved in Table R1). Results are shown here for 4 regions but are similar for the other regions.

The criterion seems quite robust as for each region shown, there is indeed an additional number of anticyclones to consider, and which is quite stable when changing this threshold, at least for regions where anticyclones are long-lived. This is less accurate for the Tel-Aviv region for example, but as this region does not accumulate anticyclones this is not really an issue.

[Figure]

**Figure R2** : Sensitivity of the total number of *order 0* anticyclones to the time percentage threshold to be considered as *order 0*. 30% means that are considered as *order 0* anticyclones dying in the studied region, **plus** the ones spending 30% of their lifetime inside it.

At last Table A2 was also modified (see Table R1 in colors) to emphasize the observed attractiveness of each regions by computing the n**et anticyclone gain** in terms of additional anticyclones : total of *importing anticyclones* minus the number of anticyclones born in the region. Due to changes performed in the scripts following the reviewer's comments, figures have differences from the ones originally submitted, which are however minor. The attractiveness of the Eratosthenes region is clear as 44 *importing* anticyclones are counted whereas only 14 were born in this region. Conversely the Haifa region seems to eject anticyclones formed here.

| Mean dynamical activity | Regions | Anticyclones born | Importing | | | | Net Anticyclone Gain | Exporting | |
| --- | --- | --- | --- | --- | --- | --- | --- | --- | --- |
| | | | Order 0 | Order 1 | Order 2 | Total | | Split | Merging |
| AC region | Eratosthenes | 14 | 23 | 19 | 2 | 44 | 30 | 7 | 0 |
| | Beyrut | 29 | 38 | 1 | 0 | 39 | 10 | 2 | 2 |
| | Tel-Aviv | 29 | 29 | 0 | 0 | 29 | 0 | 0 | 3 |
| | Nile | 53 | 52 | 2 | 0 | 54 | 1 | 6 | 1 |
| | Herodotus | 42 | 54 | 2 | 0 | 56 | 14 | 2 | 8 |
| CY region | Haifa | 24 | 18 | 1 | 0 | 19 | -5 | 0 | 6 |
| | Port-Said | 11 | 23 | 7 | 1 | 31 | 20 | 1 | 4 |

**Table R1** : for each studied region the detail of *importing* and *exporting* anticyclones, with orders. Net anticyclone gain is the difference between total of *importing* anticyclones and the ones born in the region. It should be noticed that numbers shown in column *order 0* corresponds to ordinates in Fig R2 at for a lifetime criterion of 50%.

**Specific answers :**

Abstract Line 9: "similar surface signatures correspond to very different physical properties." Is this sentence referring to the comparison between Eratosthenes and Tel-Aviv anticyclones? From fig. 12 even their surface signature seems quite different to me, the only similarity I see is being two anticyclones.

Abstract Line 9 actually refers in both campaigns to the difference between the "Tel-Aviv" and the "Eratosthenes" anticyclones. For both events, the "Tel-Aviv" anticyclone has a more intense surface signature (in terms of Rossby number, shown by a larger dot) than the "Eratosthenes" one. But the latter hides a stronger and deeper density anomaly. This sentence was changed in the manuscript.

Line 65: which poles?

Changed : Distinct regions of more intense eddying activity

Lines 123-124: can you further clarify how the colocalization of new profiles was performed through maximum velocity contours? Is it a matter of profiles' time and space with respect to the velocity contours ?

Each in situ oceanographic profile (Argo, CTD or XBT) is linked to an eddy observation - the position of a given eddy at a given day - if it falls within the maximal speed contour detected by AMEDA for this observation.

Lines 126-127: climatological background procedure not very clear to me. Please also mention the climatology usage and aim within the analysis.

Profiles falling outside any eddy contour on the day they are cast can then be considered as *outside-eddy* and representative of the "background", i.e. the climatological hydrographic properties without the eddy-induced anomaly. This background is computed as the horizontal average of all outside-eddy profiles in a spatio-temporal window : over a 150km radius around the profile location, and 30 days before and after the profile time (60 days) averaged over the years 2000 to 2018. This method then averages lots of profiles to compute the background without being blurred by seasonal variations. Backgrounds with less than 30 outside-eddy profiles are considered as not statistically significant.

This is the original DYNED-Atlas method (DYNED., 2019), the only difference being an increased number of oceanographic observation, and computed in situ temperature instead of potential to match up with XBT profiles. A paragraph was added in the Discussion section of the manuscript, concerning interannual variability biases on the calculation of the background, considering recent hydrographic studies from Ozer et al (2016).

The eddy-induced anomaly can thereby be computed as the difference between a profile inside it and the associated background of the same profile. The background is then a climatological reference specific for each profile. This result is shown in Fig. 11 of the manuscript, where we selected only profiles closer than 30km to the eddy center.

Eddy vertical content estimation is commonly performed in oceanographic campaigns, in particular for our region of interest by Moutin and Prieur (2012), but often comparing only a profile inside and another one outside the eddy. The method of reference hydrographic background, proposed in the DYNED-Atlas and used here, is intended to have a better statistical significance.

If interested, eddy vertical anomalies are available online : https://dyned.cls.fr/seewater/ , for the original 2000-2017 version (then using only Argo profiles)

Section 4.1 (and through the text): eddy exchange stands for advection from one area to another?

Eddy *exchanges* can indeed be misleading and is consecutively modified in *transfer* in the manuscript.

Line 171: From fig.3 looks like it includes also eddies generated into the region itself?

This is the definition used for *order 0 importing* eddy : "eddies dying or spending more than half of their lifetime inside the perimeter of the region." Indeed it includes eddies already formed in the region.

Line 180: case (2) what happens if an order 1 eddy dies while merging with an external one?

An *order 1* eddy is defined as an eddy which dies while merging with an *order 0* eddy (and the same applies for *order 2* merging with *order 1*). Merging with an *external* eddy cannot happen by definition.

Figures 1 and 4 (also relative caption and paragraph): I found different notations through the manuscript. Is it MDT (Mean Dynamic Topography) or the time mean of the Absolute Dynamic topography (ADT)?

The 2000-2018 time average of the Absolute Dynamic Topography is considered as the Mean Dynamic Topography.

• Lines 186-187: what if an imported eddy exits the domain (either keeping its identity or being split/merged)? Why it should not be counted as well as

exported? Maybe the eddy is just in transit within the region. I would be curious to see the complete eddy tracks dataset superimposed on the area of interest. In Fig. 3 the number of exported eddies is very low compared to the imported ones, but probably because the eddies entering the domain are excluded from the count of the exiting ones?

- Line 200: I have some doubts on the strategy of imported/exported eddies definition, the authors state "an imported cannot be also defined as exported". It probably induces very low exported eddies count, and I would revise how the attractor definition apply to the investigated areas.

- Sections 4.4-4.7: I would revisit this part according to the general comment and the comment above (Lines 186-187).

- Discussion and conclusions: I would revisit this part according to the general comment and the comment above (Lines 186-187).

These points were hopefully answered in the general comment above, we would just add here that eddy trajectories in the Eastern Mediterranean sea are very packed, with almost no clear basin-wide β-drift trajectories as with those observed in the South Atlantic ocean. Consecutively plotting all eddy trajectories would lead to an unreadable figure.

Table 2: in the transition table, considering each line, the number of anticyclones born is not equal to the sum of termination region counts. Why? Is it because of splitting?

The initial table was modified (see Table R1 above) as it was indeed not intuitive on this point. Eddy counts are not conservative, as the *order 0, order 1*, etc labels are given with respect to a studied region. For example an eddy can then be counted as *order 0* for two separate regions, if it spends 80% of its lifetime in region 1 but eventually dies in region 2. This is not a problem as the net eddy gain is precisely measured as the difference between the *importing* eddies and the ones already born inside  it (see Fig. 8 and Table R1)

Line 409 and fig.11: considering the depth of the profiles, is it potential temperature and potential density?

In addition to Argo profiles we gathered other sources of oceanographic profiles including XBT without salinity measurement. Consecutively temperature shown in Fig. 11 is in situ temperature. See also answer to specific comment line 126-127 above.

Gathering various datasets helped to have less gaps in the observation timeseries.  For example the average anomaly in 2005 (green line in Fig. 11.c ) exists only in temperature, as only XBT were available this year.

**Technical notes :** We thank the reviewer for reporting these mistakes that were corrected.

**Reply to reviewer comment 2**

We thank the reviewer for his/her very helpful remarks, in particular regarding the physical processes at stake and the new information brought compared to mean picture.

> (1) There are in the literature a bunch of eddy detection algorithms, some of them based on lagrangian tracking (Mason et al., 2014; Conti et al., 2016; amomng others). . How data from DYNED compare with them?

Present study relies on the AMEDA algorithm, which is a mixed geometric-dynamical approach, presented by Le Vu et al (2018). It uses the velocity field to find the eddy centers in extrema of the local normalized angular momentum – similarly to Conti et al (2016) – and then looks for surrounding closed SSH contours to find the eddy contours – similarly to Chelton et al (2011) or Mason et al (2014) -.

The main improvement of the AMEDA algorithm is an effective detection of merging and splitting events (see also answer to point 5), which allowed to successively track eddy network and connectivity due to Agulhas rings drift between the 2 sides of the South Atlantic ocean by Laxenaire et al (2018). In the Mediterranean sea, this algorithm was also used to track Ierapetra eddies over several years (Ioannou et al, 2017) and Algerian eddies (Garreau et al, 2018). It was also applied in the Arabian sea (de Marez et al, 2019)

> (2) The manuscript lacks of dynamical information in order to better understand the eddy formation (frontal instability?; flow topography interaction?, etc.) The inclusion of information about MKE and EKE (or MEKE) will clarify this issue.

The scope of the article is to develop a methodology aiming to recognize statistical patterns in eddy dynamics, and is not focused on their formation. However since a significant number of eddies drifting to the Eratosthenes region are formed near the coast, we can assume that some of them are formed by instabilities of the along-shore current. Such eddy detachments form the coast were already spotted by Hamad et al (2006). But our study shows that some eddies converging towards Eratosthenes are formed westwards in the Herodotus region. A possible formation process there could be instabilities of the meandering Middle-Mediterranean Jet. Physical processes leading to eddies formation should constitute a next study, as they are important to understand water masses transported by eddies.

For discussion purpose, a mean EKE map is shown in Fig. R3. It illustrates the fact that focusing on eddy intensity overrepresents intense eddies with strong variability (Ierapetra eddies) but masks important eddy dynamics and recurrent eddy patterns. Similarity with Fig.1 in Amitai et al (2010) is striking.

**Mean Eddy Kinetic Energy, 2000-2018**

**Figure R3 :** Mean eddy kinetic energy, computed using the geostrophic speed derived from Sea Level Anomaly in AVISO products. Ierapetra eddy is a very prominent feature, but the Mersa-Matruh structure also appears, whereas the Eratosthenes region is blurred

> (3) Authors in the discussion argue that convergence of AE in the southern levantine basin towards the Eratosthenes is clear but some issues are still missing regarding the role of the long living structure in attracting eddies. Does the authors think that advecting viertual particles (from the geostrophic velocities) on advecting eddies (inside and outside its maximum radius) will provide some information about this guess?.

This issue is partly addressed in the discussion of the manuscript, whether it is the bigger Eratosthenes anticyclone that is pulling smaller ones to the seamount, or anticyclones formed in the attraction basin that are drifting to what seems to be a convergence point (similarly to particles). Or in other words : do the observed convergence occurs from eddy-eddy interaction or from eddy advection ?

It is very hard to answer this issue, however it can be seen on the Eratosthenes anticyclone histogram (Fig 7.a) that every time the bigger anticyclone constituting the attractor dies, it is quickly replaced by another one. The Eratosthenes seamount then seems to be a preferred stranding point for anticyclones. On the other hand 19 *order 1* eddies are detected in this region, most of them merging with *order 0* eddies in the Eratosthenes region. A importing flux of roughly 1 anticyclone/year can be approximated, highlighting the importance of eddy-eddy interactions.

To answered the question asked : advecting virtual particles could indeed be an idea to assess this issue.

(4) Something that would enforce the work from an oceanographic point of view is to clarify the different polarities (i.e. +1 AC -1 CE) found in the different areas identified.

For each studied region within our area of interest, we attributed an averaged dynamical activity inferred from the MDT (see Fig.4 in the manuscript or Table R1). However, it appears that (anti)cyclones are not always found in averaged (anti)cyclonic regions. As reported in Table R1, Haïfa region, although being clearly cyclonic on average on the MDT, hosts the formation of 24 anticyclones over 19 years.

Actually the average occurrence of an eddy of a given polarity can directly be read looking at Fig.4 of the manuscript : it shows for each pixel the time percentage spent inside the maximal speed contour of an eddy. Almost permanent anticyclones such as Mersa-Matruh or Eratosthenes are highlighted by a strong presence, whereas areas with intense but fluctuating eddies are less marked, such as Ierapetra or the Beyrut region. Only reading Fig. 5 we can inferred the probability to fall inside an anticyclone over the top of Eratosthenes seamount ($33.6°N$ ; $32.6°E$) 50% of the time, and almost 0% for cyclones.

The DYNED method used to compute the reference background (see reply to the first reviewer) is precisely intended to retrieve the eddy physical anomalies and take into account the various occurrences of cyclones and anticyclones.

(5) A better explanation about the tracking algorithm is also desired.

Some precisions were added to the manuscript to give details about the tracking method. Briefly, AMEDA minimizes a cost function taking into account spatial proximity but also similarity in size and Rossby number over a given correlation time to gather different eddy observations in tracks. Merging and splitting events are detected as the outcome of eddy interactions, defined as a period when 2 eddies share a closed SSH contour with averaged velocity higher than the ones for the eddies taken separately (see section 5.b in Le Vu et al , 2018)

---

## Author Response (AR2)

**Reply to 2$^{nd}$ review**

We thank the reviewer for accepting the manuscript in its revised form, and agree on the minor adjustment proposed.